# Meta-Analysis of Collaborative Inhibition Moderation by Gender, Membership, Culture, and Memory Monitoring

**DOI:** 10.3390/bs14090763

**Published:** 2024-08-31

**Authors:** Xiaochun Luo, Boyao Zhao, Weihai Tang, Qian Xiao, Xiping Liu

**Affiliations:** 1Faculty of Psychology, Tianjin Normal University, Tianjin 300387, China; 2301340039@stu.tjnu.edu.cn (X.L.); 2201340028@stu.tjnu.edu.cn (Q.X.); lxp3771@sina.com (X.L.); 2Department of Student Affairs, Sichuan College of Architectural Technology, Chengdu 610399, China; 3CAS Key Laboratory of Mental Health, Institute of Psychology, Chinese Academy of Sciences, Beijing 100101, China; zhaoby@psych.ac.cn; 4Department of Psychology, University of Chinese Academy of Sciences, Beijing 100049, China; 5College of Education Science, Henan Institute of Science and Technology, Xinxiang 453003, China

**Keywords:** collaborative inhibition, retrieval strategy disruption, retrieval inhibition, memory monitoring, transactive memory

## Abstract

Collaborative inhibition is a counterintuitive phenomenon. While the specific mechanisms through which social factors influence collaborative inhibition remain unclear, this study aims to shed light on the impact of gender, relationships, and culture in order to better understand the factors shaping collaborative inhibition. A meta-analysis was conducted to analyze subgroups of social factors, including collaborative pairing, gender, membership, and culture, as well as cognitive factors like memory monitoring. Collaborative inhibition was found to be a robust effect (*p* < 0.01), with moderating effects of pairing on gender (*p* < 0.01), membership (*p* < 0.01), culture (*p* < 0.01), and memory monitoring (*p* < 0.01). The findings indicate that collaborative inhibition is a consistent phenomenon influenced by both social and cognitive factors. Moreover, the study discovered that memory monitoring can successfully moderate collaborative inhibition, although the underlying mechanism requires further investigation.

## 1. Introduction

Collaborative memory can be defined as a process involving two or more people collaborating to remember previously experienced information [1]. The experimental examination of collaborative memory typically includes a learning phase presenting the stimuli (e.g., words) and a subsequent collaborative or individual recall task. In individual recall conditions, participants freely recall as much information as possible from the learning phase. In collaborative recall conditions, participants recall this information collaboratively [2]. Research shows that collaboration is not always beneficial for memory. A group of individuals working together (i.e., a collaborative group) remembers less than their potential, measured as the nonredundant sum of the same number of individuals working alone (i.e., a nominal group). This outcome is known as collaborative inhibition [3,4].

Basden et al. (1997) proposed the retrieval strategy disruption hypothesis and suggested that exposure to other group members’ responses during collaborative retrieval may be responsible for collaborative inhibition [5]. During encoding, collaborative group members develop their own subjective organization of the studied materials, resulting in differing optimal retrieval strategies. During collaborative remembering, when participants hear other group members recall information in an order that is inconsistent with their own retrieval strategies (e.g., our well-traveled participant hears another recalling the cities in alphabetical order), they have to change to different and less effective retrieval strategies, resulting in collaborative inhibition. As nominal group members always work alone, they are not exposed to any disruption and are free to rely on their own optimal retrieval strategies, resulting in greater recall for them than collaborative group members.

The retrieval inhibition, on the other hand, suggests that during collaborative retrieval, the retrieval of specific items by group members inhibits the representation of unretrieved items by other members. This leads to lower activation levels and difficulties in retrieving the remaining items. As a result, the group’s ability to collaborate on retrieval is reduced, leading to lower retrieval scores compared to the nominal group [6,7].

Based on the two theories, the two meta-analyses found that collaboration style, type of material, and relationship moderated collaborative inhibition [8,9], whereas category size, number of learnings/tests, and type of encoding (conscious or unconscious) did not modulate collaborative inhibition [9], supporting that retrieval inhibition and retrieval strategy disruption together influence collaborative inhibition.

Currently, explanations of collaborative inhibition focus on social and cognitive factors theoretically. Researchers suggest that collaboration inhibition may arise due to social lofting and fear of negative evaluations [10,11]. In collaboration, the dispersal of responsibilities leads to social loftiness and lower productivity. Alternatively, individuals collaborating may fear criticism and thus report only highly confident answers, reducing the number of recalls. Yet the idea is not supported by empirical evidence.

Weldon et al. (2000) discovered that collaborative inhibition remains even when group motivation is increased [11]. Similarly, Wang (2016) found that collaborative inhibition remained even when participants’ motivation and competitiveness were increased and diffusion of responsibility was reduced [12]. These studies suggest that social loafing is not the primary reason for collaborative inhibition. However, it is unclear whether collaborative inhibition varies based on factors such as gender, membership, and cultural conditions.

### 1.1. Gender

Previous studies have only compared the relationship between gender and collaborative inhibition in same-sex collaborative conditions (e.g., male-male, female-female). No opposite-sex-paired studies have been conducted. Weldon et al., (2000) suppose that females may be more hesitant to cooperate in opposite-sex pairings than in same-sex pairings [11]. The assumption was that collaborative inhibition would only manifest when males were paired with females. However, the study, which exclusively involved female participants, revealed that collaborative inhibition was still present. However, the shortcoming of this study is that only female participants were used, and there was no control group. The fact that there is collaborative inhibition in female-female pairings does not indicate that there is also collaborative inhibition in male-female pairings, which needs to be further investigated.

Meanwhile, Andersson (2001) found no significant differences between males and females in terms of collaborative inhibition [13]. The study employed a design that grouped female-female pairs and male-male pairs to investigate the impact of gender on collaborative inhibition through spatial and verbal tasks. The results indicated that while females outperformed males in recall scores for the verbal task, there was no significant difference observed between the genders in terms of collaborative inhibition. This study primarily focused on the effects of gender composition rather than the interactions between genders.

Gender interaction offers a compelling lens for analysis; men and women may process information differently, and when paired for recall tasks, the variance in organizational approaches could lead to distinct retrieval strategies, resulting in varying levels of collaborative inhibition. The disparities observed in male-female pairs might stem from differences in group memory dynamics between the genders, whereas the collaborative inhibition within same-gender pairs—either female-female or male-male—could be attributed to individual memory differences. To delve deeper into this phenomenon, the current study considered gender as a moderating factor for collaborative inhibition, aiming to explore its influence on the collaborative process.

### 1.2. Membership

Studies on collaborative memberships have targeted various types of relationships, including couples, friends, acquaintances, and strangers. Findings indicate that older couples [14] and young couples [15] do not experience collaborative inhibition. In friendships, there is a stable inhibition [13,15]; however, the level of collaborative inhibition varies depending on the relationship between the members [15]. Studies comparing acquaintances and strangers have produced conflicting results, some indicating a difference [16] and others suggesting no difference [17]. While membership affects collaborative inhibition, it remains unclear whether this effect changes with relationship proximity.

### 1.3. Cultural Research

In Western cultures, individualism is emphasized, while in Eastern cultures, collectivism prevails. Individualistic cultures prioritize independence, uniqueness, and free choice, while collectivistic cultures prioritize interdependence, social embeddedness, and loyalty to in-groups, such as family [18]. Cross-cultural studies have shown that Western participants tend to use categorical strategic organization more often when completing memory tasks compared to East Asian participants [19]. In memory tasks, Chinese participants used category organization strategies less frequently than American participants did. The differences in organizational strategies due to cultural differences may impact collaborative inhibition, but there is a lack of empirical studies on this.

### 1.4. Memory Monitoring Research

In addition to the above factors, in recent years, some researchers have proposed memory monitoring as an important influence on collaborative inhibition [16,20,21,22]. Memory monitoring refers to subjective judgments of object memory activities, including ease of learning, judgments of learning, feelings of knowing, and confidence judgments [23]. Some researchers combined memory monitoring with a collaborative memory research paradigm to explore the effects of memory monitoring on collaborative inhibition and found that collaborative inhibition disappeared when judgments of learning were added after encoding [20,21]. The collaborative inhibition phenomenon also disappeared when ease of learning judgments was added after encoding [16]. The above studies suggest that memory monitoring may be an important factor in regulating collaborative inhibition, but a comprehensive analysis is lacking.

### 1.5. Existing Meta-Analyses

It should be noted that the meta-analysis conducted by Marion & Thorley (2016) on collaborative inhibition used literature from before 2014, which is more than 10 years old [9]. Sun et al. (2023) conducted a meta-analysis on collaborative retrieval [8]. They investigated the robustness of error pruning in collaborative memory and examined the potential mechanisms and moderators. An exploration that also encompassed the phenomenon of collaborative inhibition. As the focal point was the analysis of error pruning, their selection criteria prioritized studies whose error pruning is reported or can be calculated. Consequently, studies that exclusively documented correct memory without addressing error pruning were excluded. In that research, they scrutinized the influence of three moderating variables on collaborative inhibition: collaboration style, material type, and relationship familiarity. Their analysis predominantly concentrated on cognitive elements, with the examination of relationship types being confined to familiarity and strangers. This article differs from the previous two in four principal aspects: firstly, our review is more exhaustive, incorporating both published literature and unpublished dissertations; secondly, it places a greater emphasis on scrutinizing the social factors that impinge upon collaborative inhibition; thirdly, it provides a more nuanced categorization of relationship types; and fourthly, it introduces an analysis of the role of memory monitoring in collaborative inhibition.

### 1.6. Current Study

The previous studies have certain limitations that necessitate addressing: (1) Controlled studies comparing heterosexual pairs with same-sex pairs are needed. (2) The understanding of the effect of relationships with varying levels of intimacy on collaborative inhibition remains unclear. (3) Further research is required to investigate the impact of culture on collaborative inhibition. (4) There is a need to study the role of metamemory in collaborative inhibition.

In order to address the aforementioned limitations, this study will employ a meta-analysis approach to examine the effects of gender pairing, member relationships, culture, and metamemory monitoring on collaborative inhibition. (1) The analysis will focus on two types of collaboration pairs: heterosexual collaboration groups (male-female) and same-sex collaboration groups (male-male, female-female). (2) The study will explore the influence of interpersonal distance on collaborative performance. Membership will be considered based on the Transactive Memory System Theory. According to Transactive Memory System Theory, groups may develop an implicit structure in their relationships to remember important events or tasks, as well as special techniques and cues to assist each other. Once this structure is established, participants have developed a transactive memory that extends beyond individual memory. Transactive memories are present in close relationships, such as those between couples, lovers, and friends [24]. Therefore, the relationships will be categorized into four groups: couples, friends, acquaintances, and strangers. (3) Culture will also be taken into consideration and divided into Eastern culture and Western culture. This study will categorize countries or regions based on whether they belong to an individualistic culture or a collectivistic culture. Western cultures include the U.S.A., Australia, Canada, and Germany. France Portugal Holland Sweden Switzerland Turkey, while China, Hong Kong China, and Japan fall under Eastern cultures, according to previous research [25]. (4). The analysis will encompass all papers that incorporate metamemory monitoring during collaborative memory.

## 2. Methods

We performed meta-analyses according to the Preferred Reporting Items for Systematic Reviews and Meta-Analyses (PRISMA) statement of Moher et al. (2009) and their 27-item checklist [26].

### 2.1. Data Sources and Searches

The following keywords were searched: “collaborative inhibition”, “collaborative memory”, “cooperative memory”, “collaborative retrieval”, “collaborative recall”, “collaborative remembering”, “group memory”, “group remembering”, “joint memory”, and “joint remembering” on three platforms: China National Knowledge Infrastructure, Wanfang Data, and Cqvip in Chinese languages. In addition, we searched the Web of Science, Science Direct, EBSCO, and ProQuest (dissertation) to identify relevant studies in English and other languages. The search deadline was 26 November 2023, and to avoid any omissions, references to the retrieved articles were also checked to identify further relevant studies. After preliminary screening and elimination of duplicates, a total of 76 articles in Chinese and 212 articles in English were collected.

### 2.2. Study Selection

The literature was conducted using Zotero software (6.0), and the following criteria were applied during screening: (1) studies that were not empirical, such as reviews and meta-analyses, were excluded; (2) both collaborative and nominal groups were required to report correct recall/correct recall rates; and (3) if dissertations and journal papers by the same author contained the same data, only the journal papers were included. After screening, a total of 77 articles were finally selected, which included 15 master’s and doctoral dissertations, 62 journal papers, 21 Chinese literature, 55 English literature, and 1 Portuguese literature. The selected articles comprised 140 independent studies, 185 effect sizes, and a total sample size of 19,492. The time span of the literature search was from 1996 to 2023. For a visual representation of the literature search and screening process, please refer to Figure 1.

### 2.3. Coding of Moderator Variables

There are four potential moderator variables relating to study design characteristics: (collaborative pairing, gender, member relationship, culture, and memory monitoring). The first author coded the variables for all studies, as checked by the third author.

Collaborative pairing gender was coded into two categories: (1) same-sex collaboration and (2) opposite-sex collaboration. Collaborative groups consisting of 2 males or 2 females will be coded as same-sex collaboration, and collaborative groups consisting of 1 male and 1 female will be coded as opposite-sex collaboration. To provide a clearer account of the moderating influence of gender pairing and to effectively control for potential confounding variables, the current study exclusively incorporated dyadic collaborations for the analysis of the moderating impact of gender pairing.

Member relationships were coded into four categories: (1) couples, (2) friends, (3) acquaintances, and (4) strangers. Couples or lovers will be coded as partners. Those who consider each other friends will be coded as friends. Relationships that fall between strangers and friends, such as acquaintances or classmates, will be coded as acquaintances. Those who do not know each other will be coded as strangers.

Culture was coded into two categories: (1) Eastern culture and (2) Western culture. Referring to the research of predecessors [25], The categorization of cultures as Eastern or Western is rooted in both geographical distinctions and contrasting values such as collectivism and individualism. This dichotomy reflects not only the physical separations but also the ideological and societal norms that shape the cultural landscapes of these regions. In this study, the participant groups from the following countries (United States, Australia, Canada, Germany, France, Portugal, the Netherlands, Sweden, Switzerland, and Turkey) are coded as Western culture, while the participant groups from the following countries (China and Japan) are coded as Eastern culture.

Memory monitoring was coded into two categories: (1) no, and (2) yes. During the coding process, studies that included memory monitoring (such as judgments of difficulty and judgments of learning) will be coded as “yes”, while those that did not incorporate memory monitoring will be coded as “no”.

### 2.4. Data Extraction and Quality Assessment

We used a data extraction sheet (See Appendix A) to extract the following data from the included studies: the author’s name, time of publication, subcategory, number of collaborative groups, number of nominal groups, *Cohens’ d*, gender pairing (same-sex, opposite-sex) of collaborators, member relationships (couples, friends, acquaintances, strangers), culture (Eastern culture, Western culture), and whether memory monitoring was included in the coding process (yes, no). All effect sizes labeled as “couples” represent heterosexual couples. Each independent sample was analyzed only once, and if a document contained multiple independent samples, each sample was analyzed separately. The coding process was a collective effort by the group, and a coding manual was prepared. Two members independently analyzed the data according to the coding manual and cross-checked their results after the completion of the coding process. If there were any disagreements over the coding results, the group discussed and agreed on the final results.

A quality assessment was performed to assess the risk of bias in the RCTs. We applied the updated Cochrane Risk of Bias Assessment tool (RoB2.0) [27]). Reports were classified as having a low risk of bias, some concerns, or a high risk of bias. All quality assessments were independently conducted in the review pairs, and consensus was achieved through discussion among the whole team (see Figure 2).

### 2.5. Data Synthesis and Analysis

Comprehensive Meta-Analysis (CMA) 3.0 software was used to perform all analyses. Standardized mean differences (*Cohen’s d*) were calculated between the recollection completeness scores of the collaborative and nominal groups. The effect sizes were calculated by subtracting the mean recollection completeness score of the collaborative group from the mean recollection completeness score of the nominal group; a positive effect size indicated collaboration inhibition. During the coding process, the value of *Cohen’s d* was calculated based on the raw data, such as sample size, mean, and standard deviation, if the included literature did not report *Cohen’s d*. If the included literature did not report sample size, mean, and standard deviation, the *F* or *t* values of the raw data were converted according to the corresponding formula. Standard procedures from the Effect Size Calculation website https://www.psychometrica.de/effect_size.html (accessed on 29 December 2023) were used to perform these calculations.

To determine the applicable model for meta-analysis, a heterogeneity test is required, which involves the *Q* test and the *I*^2^ test. If the *Q* test is statistically significant (*p* < 0.05), it indicates that the studies are heterogeneous. On the other hand, *I*^2^ measures the percentage of variation due to true differences in effect sizes to the total variation. If *I*^2^ is higher than 75%, it suggests high heterogeneity and a random effects model should be chosen when the *Q* test result is significant. A fixed effects model should be chosen if the *Q* test result is not significant.

In this study, the meta-analysis *Q* test for all included studies showed that *Q*_(184)_ = 688.61, *p* < 0.001, and *I*^2^ = 73.28%. This result is close to 75% for high heterogeneity and indicates that there is heterogeneity among the effect sizes included in the study. It also suggests high heterogeneity among the studies, which supports the selection of a random effects model for the meta-analysis.

Publication bias refers to the tendency for studies with significant results to be more likely to be published, which means that the published literature may not provide a complete representation of all the research that has been carried out in a particular field. To address this issue, this study includes not only published journal articles but also unpublished theses, which helps to mitigate the impact of publication bias on the research findings. To ensure the reliability of the meta-analysis results, various methods will be used to assess the presence of publication bias, including the Funnel plot, the Classic Fail-safe N value, and the trim and fill method. The Funnel plot is a graphical method used to detect publication bias. If the graph presents a symmetrical inverted funnel shape, it indicates that publication bias is small and has less impact on the meta-analysis results. The funnel plot shows that the effect values are concentrated at the top of the graph and evenly distributed on both sides of the total effect. The loss of safety factor is the number of studies that would need to be included in order for a study to lose statistical significance. When this value is greater than 5*k* + 10 (where *k* is the number of effects), it indicates that there is no significant publication bias. In this study, the fail-safe coefficient analysis revealed that at a *p*-value of 0.05, the fail-safe coefficient was 31,401, which was much higher than the critical value of 935 (*k* = 185). This suggests that there was no significant publication bias. The trim-and-fill method is another way to assess publication bias. It assumes that publication bias causes asymmetry in the funnel plot and uses an iterative method to re-estimate the corrected effect size after trimming a portion of the studies. If the effect size does not differ significantly before and after trimming, it indicates that the publication bias is small. The cut-and-patch method of analysis found that the main effect remained significant after cutting and patching 36 studies to the left. The effect size was slightly reduced from the original 0.54 to 0.40, 95% *CI* [0.33–0.51], but it still remained significant. Overall, these results indicate that there was no significant publication bias in this study (see Figure 3).

The meta-analysis software Comprehensive Meta-Analysis (CMA) 3.0 was used to manage and analyze the data. The process of the meta-analysis was as follows: on the basis of obtaining the effect sizes of each study, firstly, the random effect model was selected and a heterogeneity test was conducted; secondly, a publication bias test was conducted; and finally, a main effect test and a moderated effect test (meta-regression analysis and subgroup analysis were conducted) were conducted.

## 3. Results

### 3.1. Main Pooled Results

The test used a random effects model, and the results indicated that there was a significant main effect of collaborative inhibition. The effect size is *Cohen’s d* = 0.54 (95% *CI:* 0.49–0.58), *Z* = 23.92, *p* < 0.01. A sensitivity analysis was conducted to evaluate the effect size of *Cohen’s d*, excluding one of the studies. The analysis revealed that the effect size of *Cohen’s d* ranged between 0.525 and 0.548. The meta-analysis results indicate that collaborative inhibition is a relatively stable effect, and the estimated effect size has a high degree of stability.

### 3.2. Subgroup Analyses

The researchers conducted subgroup analysis on categorical variables and found the following results:Collaborative pairing gender (*n*_1_ = 1261, *n*_2_ = 1280) had a significant moderating effect (*Q_B_* = 15.08, *p* < 0.001), whereas opposite-sex (*n*_1_ = 228, *n*_2_ = 227) collaboration inhibition did not have a significant effect (*p* = 0.52). (Note: *n_1_* represents the number of collaborative groups; *n*_2_ represents the number of nominal groups. the same below.)Membership (*n*_1_ = 1575, *n*_2_ = 1650) had a significant moderating effect (*Q_B_* = 17.68, *p* < 0.001), with the highest collaboration inhibition effect observed in the strangers group (*n*_1_ = 472, *n*_2_ = 481, *d* = 0.75), followed by the acquaintances group (*n*_1_ = 758, *n*_2_ = 824, *d* = 0.45). The collaboration inhibition effect was not significant in the couple group (*n*_1_ = 148, *n*_2_ = 147, *p* = 0.45) or the friends group (*n*_1_ = 197, *n*_2_ = 198, *p* = 0.35).Cultural factors (*n*_1_ = 4265, *n*_2_ = 4313) had a significant moderating effect (*Q_B_* = 11.53, *p* < 0.001), with a greater collaborative inhibition effect observed in Western culture (*n*_1_ = 4265, *n*_2_ = 4313, *d* = 0.76) than in Eastern culture (*n*_1_ = 2251, *n*_2_ = 2345, *d* = 0.47).Memory monitoring (*n*_1_ = 3773, *n*_2_ = 4313) had a significant moderating effect (*Q_B_* = 18.47, *p* < 0.001). The collaborative inhibition effect was significant without the addition of memory monitoring (*n*_1_ = 3758, *n*_2_ = 3724, *d* = 0.66, *p* < 0.001), but it disappeared after the addition of memory monitoring (*n*_1_ = 507, *n*_2_ = 589, *d* = 0.14, *p* = 0.2).

For more details, refer to Table 1.

### 3.3. Subgroup Analyses (Published Only)

To bolster the credibility of our findings, we conducted a secondary subgroup analysis, which specifically excluded unpublished dissertations from the dataset. Given that the variable of memory monitoring has been examined exclusively within the context of dissertations, it was not incorporated into this particular analysis. The outcomes of this refined analysis are presented in Table 2.

Collaborative pairing gender (*n*_1_ = 732, *n*_2_ = 729) had a significant moderating effect (*Q_B_* = 12.61, *p* < 0.001), whereas opposite-sex (*n*_1_ = 228, *n*_2_ = 227) collaboration inhibition did not have a significant effect (*p* = 0.46).Membership (*n_1_* = 976, *n_2_* = 950) had a significant moderating effect (*Q_B_* = 10.64, *p* < 0.05), with the highest collaboration inhibition effect observed in the strangers group (*n*_1_ = 645, *n*_2_ = 620, *d* = 0.7), followed by the acquaintances group (*n*_1_ = 159, *n*_2_ = 159, *d* = 0.59). The collaboration inhibition effect was not significant in the couple group (*n*_1_ = 148, *n*_2_ = 147, *p* = 0.45) and the friends group (*n*_1_ = 24, *n*_2_ = 24, *p* = 0.57).The moderating effect of cultural factors (*n*_1_ = 2751, *n*_2_ = 2703) was not significant (*Q_B_* = 2.31, *p* = 0.13).

### 3.4. Relationship between the Year of Publication and Collaborative Inhibition

Given the extensive time frame of our meta-analysis, ranging from 1997 to 2023, we used meta-regression to analyze the relationship between the year of publication and collaboration inhibition. Our analysis revealed a significant moderate effect of the year of publication, *b* = −0.02, 95% *CI* [−0.04,−0.007], *p* < 0.01. These findings indicate a temporal trend where the effect of collaborative inhibition has diminished as publication time has progressed.

## 4. Discussion

### 4.1. Main Effects of Collaborative Inhibition

This study reviewed 185 studies and discovered a significant main effect of collaborative inhibition, which aligns with previous research [8,9]. The findings suggest that collaborative inhibition is a consistent phenomenon.

### 4.2. Moderating Effects of Collaborative Inhibition

#### 4.2.1. Gender Facilitation

The impact of gender on collaborative pairing has a significant moderating effect. Same-sex collaboration has a significant collaborative inhibitory effect, while opposite-sex collaboration has no collaborative inhibitory effect. The findings of this study challenge the widely accepted retrieval strategy disruption hypothesis. According to this hypothesis, if the primary cause of collaborative inhibition is indeed the disruption of the retrieval strategy, then the degree of collaborative inhibition observed should be consistent or at least not significantly different between same-sex and opposite-sex pairings. However, our research revealed a surprising absence of collaborative inhibition in opposite-sex collaborations. This discovery implies that there may be other, yet unidentified, factors at play that significantly influence the occurrence of collaborative inhibition. supporting the hypothesis of “gender facilitation”, which refers to “the special behavioral facilitation effect of the opposite sex over the same sex for people with mature sexual awareness” [28]. Studies have shown that when heterosexuals collaborate, men are motivated to perform better by courtship motives. Furthermore, heterosexual collaborations tend to be more productive and produce more output [29]. Collaborations between heterosexual couples also have advantages in interpersonal interaction and cooperation, which can aid in creative problem-solving [30]. During the collaborative process, the opposite-sex pairing of collaborating partners leads to higher output at the time of extraction, with no significant difference compared to the result of the nominal group, thus causing collaborative inhibition to disappear.

It is worth noting that within collaborative pairings, there may be various types of relationships. In same-sex pairings, this includes friends, acquaintances, and strangers, while in opposite-sex pairings, it includes couples, acquaintances, and strangers. There may be an interaction effect between relationship type and gender. However, due to the limited number of studies reporting on relationship types and even fewer reports under opposite-sex pairings regarding different relationships, this study did not conduct an analysis of interaction effects. The analysis was conducted solely from the perspective of gender, examining its impact on collaborative inhibition. Gender facilitation may exhibit varying effects across different types of relationships, such as between couples, friends, acquaintances, and strangers. Further research is required to investigate whether and how this mechanism is equally applicable to all these relationship dynamics.

The current study’s findings offer insights that may elucidate previous research outcomes that did not detect collaborative inhibition in dyadic collaborative groups [31,32]. This absence could potentially be attributed to the influence of gender pairing, which may neutralize the effect of collaborative inhibition. On another note, Marion and Thorley (2016) report in their meta-analysis that collaborative inhibition is significant in dyadic collaboration, with the caveat that it has a significantly weaker collaborative inhibition effect than triads [9]. This observation hints at a possible interplay between gender pairing and group composition, both of which might jointly influence the phenomenon of collaborative inhibition. However, it is important to acknowledge that this is purely speculative at this stage. Further empirical research is warranted to ascertain the existence and nature of any interaction between gender pairing and group size on collaborative inhibition.

#### 4.2.2. Transactive Memory System

The influence of membership has a significant impact on collaboration. The collaboration inhibition effect disappears when couples and friends collaborate, but when strangers and acquaintances work together, the collaboration inhibition effect still exists. This may be due to the role of the transactive memory system, which develops among members of stable groups to encode, store, and retrieve information shared among group members [24]. Members of this transactive memory system share information and specialize with each other. It has been demonstrated that couples and friends who have been in a relationship for a long time know each other better and possess a better understanding of each other. When recalling collaboratively, they form an internal division of labor memory by default, with each one remembering their own expertise. The use of the transactive memory system also proves effective in attenuating collaboration inhibition [33,34].

Interactions between individuals in work groups lead to the emergence of transactive memory. This happens when members learn about each other’s areas of expertise and implicitly or explicitly assign responsibility for acquiring and encoding expertise-related information to the most suitable expert [35]. Studies have found that teams composed of members who trust each other and are familiar with each other’s expertise are more effective than teams of strangers. This is because team members who have worked together for a long time can form a transactive memory system. Each member has specific areas of expertise and knows the expertise of other team members [36]. Based on this knowledge, they can effectively divide the workload. The research suggests that couples and friends who are more familiar with each other may already have a transactive memory system. However, acquaintances, such as ordinary classmates or those who barely know each other, may not have formed such a system yet. The lack of a transactive memory system can lead to collaboration inhibition, as they are subject to interference from each other’s strategies, just like strangers.

### 4.3. Culture

This study is the first to explore how cultural factors impact collaborative inhibition. This study finds that cultural factors have a significant moderating effect on collaborative inhibition, which is stable in both Eastern and Western cultures. However, the amount of collaborative inhibition effect is greater in Western culture than in Eastern culture. On one hand, the observed differences can be elucidated by the retrieval strategy disruption hypothesis. Cross-cultural studies have consistently indicated that Western participants exhibit a greater propensity for employing categorical strategies in memory tasks as opposed to their East Asian counterparts [19]. Specifically, Gutchess et al. (2006) revealed that Chinese participants were less inclined to utilize categorical organization strategies in memory tasks compared to their American peers [37]. This heightened reliance on categorization among Western participants leads to a broader array of potential categorizations for each individual, which in turn results in more pronounced mutual undermining during the retrieval process. In contrast, the more restrained use of categorization strategies by Eastern participants, coupled with their less structured approach to organization, leads to a lesser degree of undermining. This divergence in the application of categorization strategies may account for the heightened collaborative inhibition observed in Western participants as opposed to their Eastern counterparts. On the other hand, this difference may be due to the fact that Eastern culture tends to focus on collectivism and group harmony [19,38], and therefore, individuals from this culture are more likely to contribute to the group and show greater motivation to achieve, leading to a weaker effect of collaborative inhibition. This aligns with previous research indicating that motivational factors play a role in collaborative inhibition and can influence its effect size, but cannot change the phenomenon of collaborative inhibition [11]. These findings suggest that social and motivational factors should be taken into account when studying collaborative inhibition. Certainly, retrieval inhibition could indeed be one of the contributing factors to cultural differences, potentially linked to the disparities in memory styles between Eastern and Western cultures. However, the precise mechanisms behind these differences necessitate further investigation. The exploration of this phenomenon requires a nuanced understanding of cultural nuances and their impact on cognitive processes. To delve deeper into this area, additional empirical research is essential to provide a solid foundation for the theoretical framework surrounding retrieval inhibition and its role in cultural variations.

Despite the observed differences in Eastern and Western cultures revealed by this study, the binary approach still presents several limitations, such as Overgeneralization: This approach obscures the intricate diversity and regional complexities, neglecting the unique historical and social dynamics at play. Internal Diversity Overlooked: The binary model often sidesteps the rich variations within cultural spheres, such as the divergences among East Asian, South Asian, and Middle Eastern cultures. Individual Variability Ignored: Such classifications tend to disregard the significant individual differences in values, beliefs, and behaviors within a given cultural milieu. While acknowledging observed differences between Eastern and Western cultures, it is crucial to delve deeper into the causes.

Intriguingly, the exclusion of unpublished data rendered the moderating effect of cultural factors statistically insignificant, suggesting that collaborative inhibition manifests with remarkable consistency across both Western and Eastern cultures. This finding could be attributed to several factors. Firstly, publication bias may be at play, where studies reporting collaborative inhibition are more likely to be disseminated in the public domain. Secondly, it is conceivable that unpublished studies concentrated on interventions aimed at mitigating collaborative inhibition, potentially explaining the reduced or nullified presence of the collaborative inhibition phenomenon in Eastern cultures. Lastly, the observed cultural differences might be highly sensitive to outliers, indicating that the stability of cultural distinctions requires further scrutiny. The instability of these cultural disparities warrants deeper contemplation and is an area for future research.

### 4.4. Memory Monitoring

The findings of this study suggest that collaborative inhibition can be mitigated by incorporating memory monitoring, as revealed through meta-analysis. Memory monitoring involves learners assessing the difficulty and processing extent of forming object memories, their ability to achieve memory goals, and retrieval situations across different stages of information acceptance, aiding in understanding their own memory degree [39]. Previous studies have shown that collaborative inhibition disappears when EOL or JOL judgments are made before the collaborative memory is formed during the learning encoding stage [16,20,21,22]. The disappearance of collaborative inhibition in memory-monitoring conditions may stem from item-specific processing, which captures the unique encoding of information for each event [40]. The act of making judgments of learning (JOL) is thought to trigger this focused processing, as it directs attention to the distinctive features of individual memory items, enhancing their distinctive encoding [41]. This process is akin to an additional encoding task that deepens the processing and solidifies the structure of memory, potentially neutralizing the disruptive effects of collaborative recall [40]. Thus, memory monitoring could mitigate collaborative inhibition by fostering a more robust and specific encoding that resists interference. Further, refined empirical investigation is warranted to substantiate this hypothesis and explore its implications for collaborative learning practices.

Our investigation reveals a significant moderating effect of memory monitoring on collaborative inhibition; however, the interpretation of this outcome merits circumspection. Firstly, the studies supporting this effect are unpublished dissertations, which, being unreviewed, may not provide findings of sufficient stability. Secondly, there is a potential for confounding variables, the variability in subject age, and the diverse gender compositions within collaborative groups, which could compromise the purity of the findings. To circumvent the confounding impact of age, the present study conducted a sophisticated analysis of the moderating influence of age on collaborative inhibition. By amalgamating the extant data, age was meticulously categorized into four developmental epochs: childhood (4.34–11.01 years, *k* = 14), adolescence (13.22–17.09 years, *k* = 16), adulthood (18.41–28 years, *k* = 75), and old age (67.2–79.15 years, *k* = 11). This stratification facilitated a meticulous examination of age as a potential moderator in the context of collaborative inhibition. The meta-analytic results underscored the significance of age as a moderator (*Q_B_* = 11.36, *p* < 0.001). Intriguingly, within the elderly cohort, the effect of collaborative inhibition was not statistically significant (*d* = −0.09, *p* = 0.68). Conversely, the adult (*d* = 0.65), adolescent (*d* = 0.44), and childhood (*d* = 0.52) groups exhibited significant collaborative inhibition effects (*p*s < 0.01). The findings collectively indicate a consistent pattern of collaborative inhibition across the child, adolescent, and adult age brackets. This consistency implies that, within the metamemory monitoring group, which encompasses a broad age range from 8.7 to 20.46 years, age does not emerge as the principal determinant of collaborative inhibition. It is the manipulation of memory monitoring, rather than age per se, that is implicated in the mitigation of collaborative inhibition within this group.

Despite these imprecisions, the implications of memory monitoring’s impact on collaborative inhibition are noteworthy. If the introduction of memory monitoring into collaborative tasks can counteract inhibition, it could greatly facilitate improvements in the efficiency and accuracy of collaborative work. This is especially relevant for collective eyewitness recall and enhancing accuracy in small-group learning environments. While our findings are preliminary, they suggest a meaningful direction for future research. A more rigorous approach, including stringent control of confounding variables and reliance on peer-reviewed evidence, will be imperative to validate and expand upon these initial insights.

### 4.5. Study Limitations and Future Directions

The current study examines collaborative inhibition from a meta-analytical perspective and provides a detailed discussion of its mechanisms and influencing factors. However, the study has some limitations: (1) Due to limited research, direct evidence validating the retrieval inhibition hypothesis is lacking. With only one quantitative study available, it is challenging to confirm the role of retrieval inhibition in collaborative inhibition. (2) The sample size is limited, with few detailed studies on collaborative pairing by gender or membership, leading to an insufficiently robust sample. According to our review, the synthesis of existing literature yields no universal standard for the optimal count of effect sizes in meta-analysis, with opinions ranging from a minimal threshold of two studies for foundational analysis [42] to a stringent minimum of five effect sizes per categorical moderator variable level for enhanced analytical rigor [43]. The strategic employment of moderator variables has been posited to alleviate the demand for an extensive array of effect sizes [44]. The succinct effect size set offers pragmatic advantages, especially within specialized research niches where data are scarce, thereby facilitating meta-analytic feasibility and serving as a crucible for deeper investigative pursuits. Nevertheless, this parsimony incurs drawbacks, such as diminished statistical power that may eclipse the detection of substantive effects and escalate the precariousness of effect size estimations. The limited scope may also fail to encapsulate the heterogeneity spectrum, risking a misrepresentation of the broader field and susceptibility to publication bias that could distort the meta-analytic generalization towards significant outcomes. In essence, our study, albeit modest in effect size, contributes a significant perspective on the modulation of social factors within the realm of collaboration inhibition, highlighting the merit of targeted meta-analytic inquiry. (3) In this study, the inclusion of unpublished dissertations may have implications for the internal validity of our results. The debate over incorporating dissertations into our meta-analysis was robust, with acknowledged drawbacks including potential quality control deficiencies due to the absence of peer review, concerns over data integrity and transparency, and the inherent accessibility challenges of unpublished works. Conversely, the merits of including dissertations are noted: firstly, they enhance the breadth and diversity of our study sample by addressing research areas or methodologies not covered in published literature. Notably, the moderator variable of memory monitoring in our study was primarily derived from dissertations, underscoring their unique contribution. Secondly, their inclusion mitigates publication bias, integrating results that may diverge from prevailing trends and thus broadening the representativeness of our meta-analysis [45]. After a thorough evaluation of these factors, our research team resolved to include dissertations to ensure a comprehensive and robust meta-analysis. We conducted rigorous quality assessments of all included studies and proactively engaged with dissertation authors to obtain more comprehensive datasets, thereby upholding the integrity and reliability of our research.

In the future, there is scope for further research on the performance of collaborative inhibition in contextual memory and its mechanisms. Contextual memory comprises item memory and source memory. This study focuses on the performance of collaborative inhibition in item memory. The mechanism of collaborative inhibition in source memory can be analyzed in future studies. Collaborative memory has its advantages and disadvantages. The disadvantage is collaborative inhibition, while the advantages include error pruning (reduced incorrect recall in collaborative groups compared to nominal groups) and post-collaborative facilitation (higher correct rate in collaborative group individual extraction compared to second individual retrieval in nominal groups). In future studies, the mechanisms of error pruning and post-collaborative facilitation can be deeply analyzed, as well as the common mechanisms behind collaborative inhibition, error pruning, and post-collaborative facilitation.

### 4.6. The Year of Publication and Collaborative Inhibition

This study found a decline in the collaborative inhibition effect in the sample as the year of publication increased. This discovery may shed light on the research trajectory of collaborative inhibition to some degree. Initially, the focus lies on uncovering the phenomenon of collaborative inhibition [5,46]. Subsequently, the investigation shifts to identifying the factors that influence this phenomenon [47,48]. Following this, various strategies are employed to mitigate the effects of collaborative inhibition [2,49]. As additional variables are incrementally introduced, the impact of collaborative inhibition is observed to diminish. This progression could potentially account for the attenuation of collaborative inhibition over time. However, this remains a preliminary hypothesis, and further inquiry is warranted to elucidate the underlying reasons.

### 4.7. Innovations and Practical Significance

This study is unique as it explores how gender and cultural factors affect collaborative inhibition. It is also the first to use meta-analysis to examine the relationship between memory monitoring and collaborative inhibition. Furthermore, the study has refined the moderating effects of different membership relationships on collaborative inhibition. The data analyzed in this study spans from 1996 to 2023, making it more comprehensive and important for future research to investigate the intrinsic mechanism of collaborative inhibition in depth. Additionally, it has significant reference value for the application of collaborative memory in education and testimonial research on witnessing crimes.

This study’s practical relevance is evident in several key educational and collaborative scenarios. Gender diversity is very important in group learning. It is recommended to form mixed-gender groups for group learning in educational environments. This strategy can significantly diminish the impact of collaborative inhibition, thereby enhancing the group’s overall learning experience. The Impact of Relationship Intimacy: The study highlights that the closer the relationship between collaborators, the more effective their collaborative memory. Thus, when embarking on cooperative tasks, selecting partners with whom one shares a closer bond can lead to superior outcomes. Memory Monitoring in Collaboration: To overcome the inherent limitations of collaborative efforts, the study suggests the implementation of memory monitoring in tasks that require collaborative memory, such as witness recall in legal contexts or collective learning sessions. By integrating these insights, the research provides a framework for optimizing collaborative efficiency in various settings, ensuring that the collaborative process is both productive and reliable.

## 5. Conclusions

This study used a meta-analysis approach to analyze the effects of collaborative inhibition. The results showed that collaborative pairings of gender, membership, culture, and memory monitoring had significant moderating effects. Gender facilitation was found to be the cause of gender differences. The study also suggested that the interactive memory system can be used to explain the effect of membership on collaborative inhibition. In addition, the study highlighted the need for further investigation into the moderating mechanisms of cultural differences and memory monitoring on collaborative inhibition. It is further validated that social cognitive factors are one of the reasons influencing collaborative inhibition.

## Figures and Tables

**Figure 1 behavsci-14-00763-f001:**
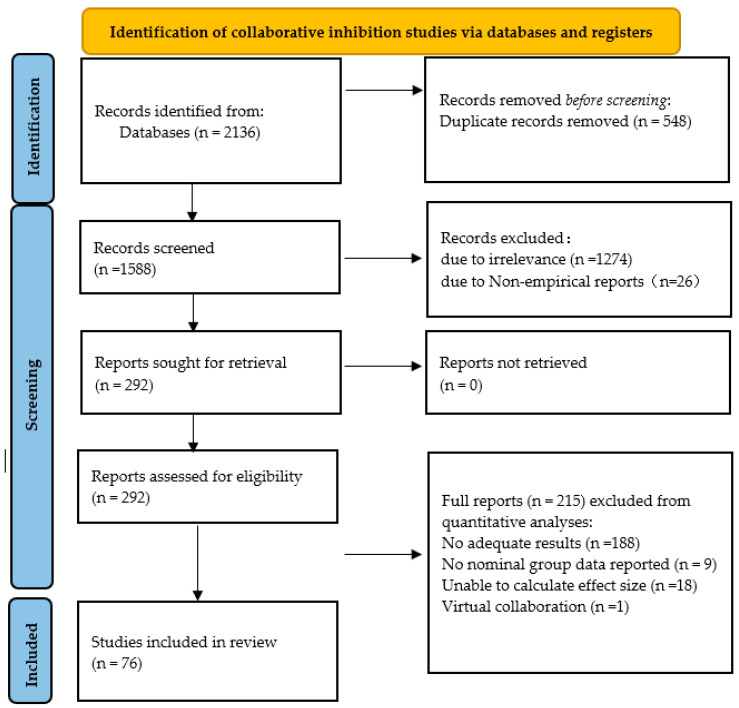
Flow diagram of the literature search and screening process.

**Figure 2 behavsci-14-00763-f002:**
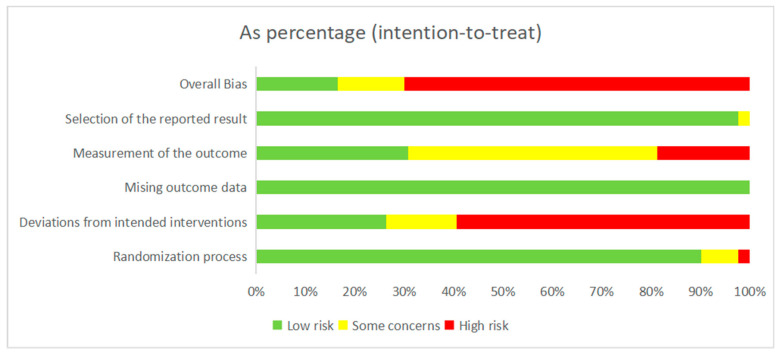
Cochrane risk of bias evaluation.

**Figure 3 behavsci-14-00763-f003:**
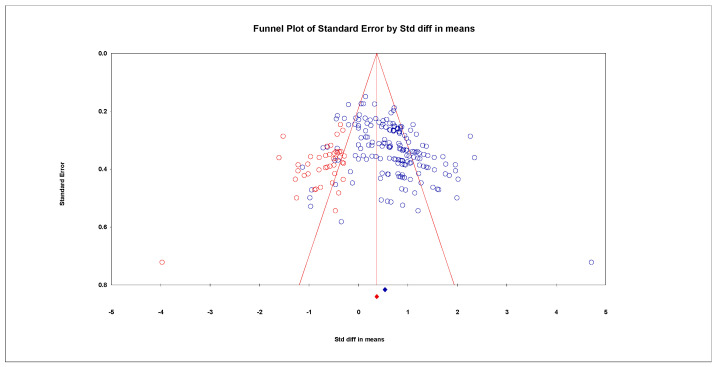
Duval and Tweedie Trim and Fill Imputed Funnel Plot. Blue circles are those studies that were included in the meta-analysis; red circles represent imputed studies that are thought to be missing due to publication bias. The blue diamond represents the original effect size, while the red diamond represents the adjusted effect size. The widths of the diamonds represent variance.

**Table 1 behavsci-14-00763-t001:** Subgroup analysis based on the dimensions.

Subgroup	Heterogeneity			Dimension	*k*	*d*	95% *CI*		Bilateral Test	
	*Q_B_*	*df*	*p*				Lower	Upper	*z*	*p*
Collaborative pairing gender	15.08	1	<0.001							
				same-sex	40	0.64	0.45	0.83	6.65	<0.001
				opposite-sex	15	−0.11	−0.44	0.22	−0.65	0.52
Membership	17.68	3	<0.001							
				couple	9	−0.19	−0.69	0.31	−0.75	0.45
				friends	8	0.18	−0.20	0.55	0.93	0.35
				acquaintances	21	0.45	0.26	0.63	4.71	<0.001
				strangers	46	0.75	0.57	0.93	8.06	<0.001
Culture	11.53	1	<0.001							
				Eastern culture	90	0.47	0.33	0.60	6.85	<0.001
				Western culture	95	0.76	0.66	0.86	14.90	<0.001
Memory monitoring	18.47	1	<0.001							
				no	173	0.66	0.57	0.75	14.24	<0.001
				yes	12	0.14	−0.08	0.36	1.27	0.20

Note: *k* represents the number of effect values; *Q_B_* represents the heterogeneity test statistic.

**Table 2 behavsci-14-00763-t002:** Subgroup analysis based on the dimensions (published only).

Subgroup	Heterogeneity			Dimension	*k*	*d*	95% *CI*		Bilateral Test	
	*Q_B_*	*df*	*p*				Lower	Upper	*z*	*p*
Collaborative pairing gender	12.61	1	<0.001							
				same-sex	24	0.70	0.40	1	4.56	<0.001
				opposite-sex	15	−0.11	−0.44	0.22	−0.65	0.52
Membership	10.64	3	<0.05							
				couple	9	−0.19	−0.69	0.31	−0.75	0.45
				friends	2	0.22	−0.53	0.97	0.57	0.57
				acquaintances	8	0.59	0.18	0.99	4.71	<0.001
				strangers	30	0.70	0.45	0.95	5.5	<0.001
Culture	2.31	1	0.13							
				Eastern culture	42	0.54	0.28	0.80	4.01	<0.001
				Western culture	93	0.76	0.66	0.86	14.62	<0.001

Note: *k* represents the number of effect values; *Q_B_* represents the heterogeneity test statistic.

## Data Availability

Data will be made available on request.

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
