# Peer review of "Meta-Analysis of Collaborative Inhibition Moderation by Gender, Membership, Culture, and Memory Monitoring"

_behavsci, 2024, doi:10.3390/bs14090763_

Round 1

Reviewer 1 Report

Comments and Suggestions for Authors

The key questions posed in this manuscript are 1) how social factors may impact the magnitude of the collaborative inhibition effect in group retrieval, and 2) how strong of an impact memory monitoring has on this effect. The authors approach this investigation using meta-analytical techniques.  While previous meta-analyses on this memory phenomenon have been conducted, they did not code the data in a way that affords looking into the above-mentioned factors. The results are consistent with previous meta-analyses demonstrating that collaborative inhibition is a robust effect, and in turn, each of their moderator analyses was found to be significant.

I find these aims to be interesting and valuable to research into social memory.  Conducting empirical studies on this topic can be pragmatically challenging. Overall, I find this topic to be important to the overarching psychological (and memory specifically) and I would like to see this report published, with several revisions.  The literature review was thorough and I felt the authors summarized it well for the length used.  In addition, the explanations of how meta-analytics operate were something I believe helped make the manuscript more accessible for readers who may not know how to interpret these types of results.

Here are areas of the manuscript I believe should be expanded on and/or clarified prior to publication:

Feedback –

1.      When it comes to the methodological approach, the authors did a good job of describing their search process.  However, important information relating to their moderator variables was not provided in a way that was explicitly defined for each variable.  A section on how each moderator was coded should be provided in the final published manuscript. 

For example, the authors state in their introduction that gender has traditionally been studied mainly as same-sex collaboration.  Thus, it is assumed the moderator is designed to give insight into whether having a heterogeneous versus homogenous gender composition impacts collaborative inhibition.  What is missing here is how precisely this was coded in the analysis.  Was this variable coded as studies that have specifically formed collaborative groups to be homogenous versus everything else?  This context is implied, but not directly stated. 

In addition, acknowledgment of this type of information comes with additional limitations that should be disclosed.  An example of one (and a question I had while reading the manuscript) was if the magnitude of the heterogeneity in a group changes the group composition in a way that hampers interpretation.  As someone who has conducted collaborative memory studies at a university setting in the United States, I know the gender distribution has often been skewed in favor of those who identify as female (as the sample pool itself is skewed).  This is relevant in that when conducting a collaborative memory study without controlling for gender composition, the degree of heterogeneity in a group is dependent on the gender distribution of the university where the study was conducted.  This contrasts with the same-sex level (i.e., homogeneous level) of the variable as it is explicitly controlled and discrete. 

In sum, the authors should be clearer on the operational definition of each moderator variable they coded and integrate these not only in their methods but also into their discussion and interpretations.

2.      Upon finishing the discussion section, I felt it lacking.  The results themselves are interesting and have merit and I found myself excited to get to their unpacking.  Yet, the interpretation of the findings was underwhelming and left much up to the reader.

The connection to the theoretical underpinnings was sparse in the discussion. In the introduction section, two prevalent cognitive hypotheses for collaboration inhibition are briefly discussed (i.e., retrieval strategy disruption and retrieval inhibition).  However, the findings were not put into the context of these theories and instead a statement indicating the findings cannot be used in the previously mentioned contexts was provided in the section on study limitations. Even though empirical studies are often how researchers approach theoretical mechanisms in cognition, the meta-analytic findings should be placed in the context of the entire body of literature, even if that context is limited.

I found myself wondering why culture was modulating the effect, an interesting finding that piqued my interest. The authors provide a statement that the finding may have to do with analytical versus holistic processing (i.e., individualism versus collectivism in this framing), which they previously described, and the possibility of differential motivation driving the modulation. Social loafing as an explanation for collaborative inhibition has been one that has generally been considered to not have much of an impact on the effect.  This is implied by the authors in their introduction and literature supporting this notion was referenced.  Thus, if motivation drives this finding, the authors should interpret why this occurs, and why motivation would circumvent the leading cognitive hypotheses relating to retrieval strategy disruption and retrieval inhibition that it contrasts.

Importantly, the discussion of the application of the findings to real-world settings was absent.  Empirical studies in this area of memory often extrapolate their findings to how memory may be impacted by everyday collaboration.  The findings the authors present are important to this extrapolation, as they add an additional layer to those interpretations.  The manuscript would benefit by emphasizing and highlighting those connections to the results.

3.      The authors do an excellent job concisely summarizing their mathematics, however I found myself wanting more clarity when it came to sample sizes of each comparison.  This was briefly mentioned in the limitation section.  The reader should be directly provided with the sample size of each level of each variable.  If these are heavily skewed, the authors should draw greater emphasis to this aspect and specifically state the sample sizes for each level of each variable in their comparisons.

4.      When it comes to culture as a variable, it is intuitive to define the variable in a dichotomous manner as much of the literature does.  However, culture is a complex social construct, and comes with additional nuances to consider and possibly disclose.  Was this variable also coded for the country?  Were there any significant differences between the countries ascribed to each level of this variable?  If so, why?  For this reason, I think the authors should include an analysis of this sort for each level of this variable - based on a dichotomous definition there should be no significant differences within that level if the operational definition is properly delineating the levels of the variable.  If they do differ significantly, it draws into question the framing as well as the interpretation of this variable.

Furthermore, the construct of culture within psychological research has been broken down into multiple dimensions, two of which are examined in this analysis. The authors have defined culture based on geographical location, and while this is appropriate, the construct of culture refers to much more (e.g. socioeconomic status, ethnicity, gender identity, and religion).  I implore the authors to re-frame this variable away from the term “culture” as an umbrella construct and instead change the framing to geographical upbringing (or something of the sort).  That is not to suggest that both gender and geographical location are not related to culture, rather they are each a specific dimension of culture.  This clarity, and framing, may help prevent overgeneralization or oversimplification of the construct being investigated.

5.      My last item to note stems from my own curiosity.  In recent years publication bias has become a growing concern.  The authors did their due diligence in this regard in terms of funnel plots (which should be included in the manuscript proper upon revision rather than in the supplements due to their importance) and fail-safe N. 

I think it may be worth exploring if the results remain consistent, or if the findings change, contingent on the sample in the analyses containing only studies that have gone through the peer-review process versus only committee assessments (as approximately 20% [15 out of 77] of the studies included did not go through the publication process).    If they are consistent, this would greatly bolster the confidence in these findings.  In the same vein, because the studies included in the analysis span many years, there is great value in seeing if a decline effect was present in the sample as the year of publication increased.

Reviewer 2 Report

Comments and Suggestions for Authors

Summary and General Evaluation:

This paper reports a meta-analysis on collaborative inhibition that includes moderating variables of gender, relationship status, culture, and memory monitoring.  The authors found a reliable effect of collaborative inhibition.  Further, they found that gender moderated the effect (same sex collaborators showed collaborative inhibition while opposite-sex collaborators did not), relationship moderated the effect (larger collaborative inhibition effects for strangers), culture moderated the effect (greater collaborative inhibition for Western samples relative to Eastern samples), and memory monitoring moderated the effect (making metacogntive memory monitoring judgements eliminated collaborative inhibition).  The authors concluded that collaborative inhibition is a robust effect that is influenced by both social and cognitive factors.    

Overall, I found the paper interesting and novel.  To my knowledge, this is the first paper to examine memory monitoring and cultural influences on the magnitude of collaborative inhibition.  However, my main concern is the small number of effect sizes included in the sub group analyses and the possibility that there may be confounding variables and interactions influencing the subgroup analyses.  As a result, I am not sure how to interpret the findings.  I outline below specific comments I hope the authors will consider as they revise and improve the paper.

Specific Comments:

    1. Please provide additional information about the number of effect sizes in each condition.  Specifically, Table 1 reports “k” as the number of effect sizes in each subgroup.   Some of these k numbers are quite small.  For example, there are only 8 effect sizes coded as “friend”, 9 coded as “couple”, 12 coded as “yes” memory monitoring, and 15 coded as “opposite-sex”.  I am concerned that there is not enough power to draw conclusions from such small samples.  Please help the reader understand the pros and cons of including such small groups in the meta-analysis.

    1. Please explain how the authors managed possible confounds with the subgroup codes.  For example, there are just 12 effect sizes in the “yes memory monitoring” group.  It appears that the 4 papers cited in the introductory section on memory monitoring (p. 3) were conducted with different age groups (these papers all have “developmental” in their title, but I could not pull the papers because they are unpublished master’s theses).  Is the finding that there is less collaborative inhibition in the memory monitoring groups confounded with less inhibition for younger age groups?  And/or are there other possible confounds?  More generally, how did the authors deal with this?

    1. Please provide additional discussion about how/if the subgroups might interact?  For example, Table 1 reports that there were 15 effect sizes in the “opposite-sex” condition and 9 effect sizes in the “couple” condition.  Were all 9 of the couples opposite sex?   Relatedly, the authors state that the mechanism underlying the moderating effect of gender is gender facilitation (p.8).  Does this apply equally to all opposite sex partners (i.e., couples, friends, acquaintances, strangers)?

    1. Please provide additional rationale for including unpublished master’s theses and doctoral dissertations in the meta-analysis. 

4 a. One concern is that at least some of these experiments are unpublished because they have experimental flaws or other issues that preclude them from being published.  How did the authors deal with internal validity issues?  I recommend mentioning internal validity issues with unpublished studies as a limitation in the General Discussion.

4 b.  Another concern is the availability of the unpublished papers.  I was interested to read several of the unpublished theses cited, but I could not easily access unpublished theses from various universities.  If there is a recommended way to access the unpublished papers, please report that in the manuscript.

    1. The authors cite a recent meta-analysis by Sun et al (2023) and state that they focused on fewer factors (page 3).  Please provide more details about the Sun et al. 2023 study and explicitly highlight how the current study is different.

Comments on the Quality of English Language

Minor edits needed to help readability

Round 2

Reviewer 1 Report

Comments and Suggestions for Authors

Excerpt From Initial Review: “The key questions posed in this manuscript are 1) how social factors may impact the magnitude of the collaborative inhibition effect in group retrieval, and 2) how strong of an impact memory monitoring has on this effect. The authors approach this investigation using meta-analytical techniques.  While previous meta-analyses on this memory phenomenon have been conducted, they did not code the data in a way that affords looking into the above-mentioned factors. The results are consistent with previous meta-analyses demonstrating collaborative inhibition is a robust effect, and in turn, each of their moderator analyses was found to be significant.”

Feedback to Authors' Response

Author Response:

“We extend our gratitude to the reviewer for the insightful and constructive feedback. In response to your comments, we have made several revisions to the manuscript:

In the Introduction section, we have now included an expanded discussion on the gender pairing to provide a clearer context for our study's focus. (The revised text can be found on page2-3, paragraph9, and line81-103.)”

Reviewer Response: This additional was thorough and I believe will be helpful to the reader.  I consider this resolved to a satisfactory level.

“The Methods section has been enhanced with a more comprehensive explanation of how the moderator variables were coded, ensuring greater transparency and replicability of our research methodology. (The revised text can be found on page6, paragraph1, and line213-237.)”

“Lastly, in the Discussion section, we have elaborated on the limitations associated with the pairing of gender as a moderating variable and added practical significance (The revised text can be found on page11, paragraph9, and line378-385.)”

Reviewer Response:  I found this addition to be very helpful.  A minor revision I would request is to give each moderator its own sub-section within the section.  This will assist the reader in quickly returning to a piece of specific moderator information after reading further. 

In addition, re-reading the gender information that was added, helped clarify for me more of the definition of the variable.  It is now salient that the collaborative groups used in the analysis (or at least this moderator) are dyadic.  After searching the manuscript, I do not see any discussion of the nuances of dyadic collaborative recall, and believe they should be disclosed.  This should be included in the introduction section of the literature review on the differences between dyadic versus triadic collaboration and how the effect is proposed to scale in magnitude with increased group size. 

Things to note in that addition, firstly, is that some studies have found no significant collaborative inhibition when using dyadic groups (e.g., Andersson & Rönnberg, 1995; Basden et al., 2000; Meudell, Hitch, & Boyle, 1995; Meudell, Hitch, & Kirby, 1992), while others have indeed found significant effects.  This may be important in the context of the gender pairing moderator and be a potential explanation as to why there has been a discrepancy in the prior literature. 

Secondly, Marion and Thorley (2016) report in their meta-analysis that collaborative inhibition is significant in dyadic collaboration with the caveat that it has a significantly weaker collaborative inhibition effect than triads. An addition relating to this should be integrated into both the introduction and the discussion as a limitation.  The reason I state this is a limitation of the analyses is dyadic collaboration is a known moderator of collaborative inhibition, and since null effects are being interpreted, it’s important to highlight the potential for an interaction between group size and group composition.

For the addition of practical significance, I am satisfied and found it to capture the spirit of my feedback.

Author Response:Thank you for your very professional opinion, based on your suggestions. We have further improved the discussion section. Firstly, in the Discussion section, we have strengthened the connection with theories and added the discussion of the cognitive factor of collaborative inhibition.”

Reviewer Response: I have reviewed this section, and I will be forward that it did not capture the spirit of my concern, and rather further exacerbated it.  It seems the authors provide a strong theoretical interpretation of their findings without fully unpacking their claim. In my opinion, this should be interpreted more cautiously due to the concern of dyadic collaboration I raise above, and the potential for interactions that may be occurring between group size (i.e., dyadic recall) and group identity homogeneity (i.e., gender pairings).  The interpretation “The findings of this study challenge the widely accepted retrieval strategy disruption hypothesis.” is directly stated and then no potential explanations are given from a cognitive perspective or in the context of that robust body of literature on the topic. In addition, the other important cognitive mechanism hypothesized to be involved during collaborative inhibition, namely retrieval inhibition (described in the introduction section), is absent from interpretations.

Secondly, in the section of Cultural Factors, we have added more discussion of the cognitive factor; (The revised text can be found on page12, paragraph4, and line438-45)1and lastly, in the section of Memory Monitoring, we have strengthened the discussion of  researchers approach theoretical mechanisms in cognition, the meta-analytic findings should be placed in the context of overarching literature to a greater extent than it was even if it is limited.  In the Discussion section, we have added practical significance.”

Reviewer Response:  Overall, I was satisfied with these additions and felt that they captured the spirit of my feedback.

Author Response: “Thank you very much for the reviewer's feedback. Indeed, there is a significant discrepancy in the sample sizes of the moderating variables in this study, primarily due to the limited number of studies on gender pairing, relationships, and metacognitive monitoring, which results in a smaller effect size. In addtitional file 1, we provide a comprehensive breakdown of the sample sizes for both the collaborative and nominal groups under each experimental condition. However, to conserve space within the main body of the text, we have refrained from detailing these sample sizes. In response to your suggestion, we have conducted an in-depth discussion and explanation in the limitations section of our study. (The revised text can be found on page14, paragraph4, and line550-567.)”

Reviewer Response: I appreciate the authors directing me to the additional file, it helps acquire the information and does disclose it.  However, as I noted in my initial review, this information is important and should be reported with each analysis to ease access to the information.

I believe this can be done clearly and concisely without adding much length of concern.  Below I provide an example:

Original (Page 9): “Collaborative pairing gender had a significant moderating effect (QB = 15.08, p < 0.001), whereas opposite-sex collaboration inhibition did not have a significant effect (p = 0.52).”

Suggested: “Collaborative pairing gender (n = 40) had a significant moderating effect (QB = 15.08, p < 0.001), whereas opposite-sex (n = 15) collaboration inhibition did not have a significant effect (p = 0.52).”

Author Response: “I am grateful for the reviewer's suggestions; indeed, culture encompasses a vast array of elements. We have had in-depth discussions regarding the adoption of the variable of "culture," and we believe that culture might be more appropriate than geographical location. The reason is that culture may more readily explain psychological influences. On the other hand, when we categorize, we often consider the culture of the subjects from this place, not just the geographical location. For example, Turkey is geographically in Asia, but Turkish culture is heavily influenced by the West, so we code the culture of Turkish subjects as Western culture. Although we have not changed the name of the "culture" variable, we fully understand the reviewer's considerations, and we have analyzed the limitations of such categorization in the discussion.

Based on your recommendations:

1. We have provided a specific explanation of cultural coding in the methods section; (The revised text can be found on page7, paragraph1, and line225-237

2. In the discussion, we have analyzed the potential drawbacks of this Eastern and Western cultural classification. (The revised text can be found on page12-13,

8 paragraph5, and line433-442.)

Reviewer Response: I enjoyed these additions; I am satisfied with them and feel they capture the spirit of my feedback. 

Author Response: “We are grateful for the insightful and professional feedback from the reviewers. In response, we have implemented the following three refinements:

Firstly, we have relocated the discussion on publication bias from the appendix to the main body of the text, the fail-safe N is detailed in the text on page 8, paragraph 2, at line 295.(The revised text can be found on page9, paragraph2, and line314-319;

Secondly, we have conducted a separate analysis by excluding effect sizes that did not incorporate unpublished data, and then we have compared these findings with those that included such data. (The revised text can be found on page10-11, paragraph2, and line346-361; page13, paragraph2, and line470-480

Finally, we have explored the correlation between the year of publication and collaborative inhibition, utilizing the year of publication as a moderating variable in our analysis. (The revised text can be found on page11, paragraph1, and line361-368; page15, paragraph3, and line585-59).”

Reviewer Response: I am happy you hear that the value of these types of analyses was acknowledged, I appreciate their addition.

Minor New Feedback:  Please replace the word “subjects” with “participants” throughout the manuscript.

Reviewer 2 Report

Comments and Suggestions for Authors

I served as Reviewer 2 on the previous version of this paper.  The authors have adequately addressed my concerns, and I found the revision much improved.

Comments on the Quality of English Language

English is generally fine-- I recommend a careful read through for minor edits

Author Response

Thank you for your suggestions, after reading the article through we have made replaced the word “subjects” with “participants” throughout the manuscript.